# Absorption of Direct Oral Anticoagulants in Cancer Patients after Gastrectomy

**DOI:** 10.3390/pharmaceutics14030662

**Published:** 2022-03-17

**Authors:** Hannah C. Puhr, Aysegül Ilhan-Mutlu, Matthias Preusser, Peter Quehenberger, Paul A. Kyrle, Sabine Eichinger, Lisbeth Eischer

**Affiliations:** 1Division of Oncology, Department of Medicine I, Medical University of Vienna, 1090 Vienna, Austria; hannah.puhr@meduniwien.ac.at (H.C.P.); aysegul.ilhan@meduniwien.ac.at (A.I.-M.); matthias.preusser@meduniwien.ac.at (M.P.); 2Comprehensive Cancer Center, Medical University of Vienna, 1090 Vienna, Austria; 3Department of Laboratory Medicine, Medical University of Vienna, 1090 Vienna, Austria; peter.quehenberger@meduniwien.ac.at; 4Division of Hematology and Hemostaseology, Department of Medicine I, Medical University of Vienna, 1090 Vienna, Austria; paul.kyrle@meduniwien.ac.at (P.A.K.); sabine.eichinger@meduniwien.ac.at (S.E.); 5Karl Landsteiner Institute of Thrombosis Research, 3100 St. Pölten, Austria

**Keywords:** cancer, gastrectomy, thrombosis, atrial fibrillation, anticoagulation

## Abstract

Direct oral anticoagulants (DOACs) are safe and effective in cancer patients treated for venous thromboembolism (VTE) or atrial fibrillation (AF). Gastrectomy is the treatment of choice in patients with localized upper gastrointestinal cancer. DOACs are absorbed in the upper gastrointestinal tract, but to what extent is unclear. In a retrospective analysis, hospital data were searched for adult patients who underwent gastrectomy for gastroesophageal or pancreatic cancer, and DOAC therapy for VTE or AF after gastrectomy. DOAC blood levels were determined by chromogenic assays before and after administration, and thromboembolic and bleeding complications were recorded. Eleven patients (median age 76 years) received a factor Xa inhibitor (FXaI; apixaban (3), edoxaban (3), rivaroxaban (4)) or the factor IIa inhibitor dabigatran (1) for VTE (7) or AF (4) after gastrectomy. Eight patients on FXaI had anti-Xa (aXa) trough levels within the expected range (ER). In all of them, aXa levels increased upon DOAC administration. Two patients on 30 mg edoxaban had low aXa trough levels. Administration of 20 mg of rivaroxaban resulted in trough levels in the ER in one of them. None of the FXaI patients had thromboembolism, while two experienced bleeding (arterial puncture site, gastrointestinal). One dabigatran AF patient with trough and peak concentrations below the ER had strokes during 110 mg and 150 mg dabigatran administration. While on apixaban, aXa levels were in the ER, and no clinical complications occurred. DOACs, particularly FXaI, were adequately absorbed in cancer patients after gastrectomy. Our observation of recurrent thromboembolic events in a patient treated with dabigatran warrants cautious use in this specific patient population.

## 1. Introduction

Thromboembolic events are prevalent in cancer patients [1], and are the second leading cause of death in this population [2]. Patients with malignancies are likewise at an increased bleeding risk, which renders effective yet safe antithrombotic strategies challenging. Direct oral anticoagulants (DOACs), including the factor Xa inhibitors apixaban, edoxaban and rivaroxaban, as well as the factor IIa inhibitor dabigatran, are licensed for the treatment of venous thromboembolism (VTE) and for stroke prevention in nonvalvular atrial fibrillation. Recent studies demonstrated effectiveness and safety of apixaban, edoxaban, and rivaroxaban in patients with cancer-associated VTE as well [3,4,5]. The use of DOACs has been recommended in cancer patients with nonvalvular atrial fibrillation [6].

DOACs have the advantage of oral administration, few drug–drug interactions, and a short half-life, which also makes them appealing in the cancer population [7]. Patients with gastroesophageal and pancreatic cancer are at particularly high risk of venous and arterial thrombosis [8,9], as well as atrial fibrillation [10]. Resection of the tumor by total or partial gastrectomy is the mainstay for achieving disease cure in patients with localized disease [11]. Since DOACs are absorbed predominantly in the upper gastrointestinal tract, treating patients with DOACs after gastrectomy raises concerns regarding achievement of adequate drug levels. Some smaller studies and case reports addressed the absorption of DOAC in patients after bariatric surgery [12,13,14,15,16,17,18], but to our knowledge, only one patient taking DOACs after a cancer-related gastrectomy has been reported [19].

It was therefore the aim of our study to evaluate the absorption of oral direct factor Xa and a factor IIa inhibitor in cancer patients after gastrectomy. We measured DOAC concentrations in plasma at trough and peak and compared them with therapeutic ranges obtained from clinical studies [20,21].

## 2. Material and Methods

### 2.1. Study Design and Patients 

In a single-center retrospective analysis, we searched the hospital record data and medical files between 2018 and 2021 and identified 11 patients who fulfilled the following criteria: age greater than 18 years; gastrectomy for upper gastrointestinal cancer; and treatment with a DOAC for VTE or atrial fibrillation after gastrectomy. Patients were managed at the Division of Oncology, Dept. of Medicine I, Medical University of Vienna, Austria, and were referred for DOAC level measurement to the coagulation clinic of the Division of Hematology and Hemostaseology, Dept. of Medicine I, Medical University of Vienna. We obtained laboratory and clinical data including patient demographics, therapeutic regimens, and clinical outcomes from hospital record data. All venous or arterial thromboembolic events, as well as bleeding episodes during the time from gastrectomy to the last available information in the medical charts, to discontinuation of anticoagulation or death, were recorded. The date of data cut-off was 5 November 2021. 

The study was approved by the Ethics Committee of the Medical University of Vienna (Ethics Committee Approval Number 1896/2021, 3 November 2021). 

### 2.2. Coagulation Testing and DOAC Concentration Measurement 

The prothrombin time-based international normalized ratio (INR) and the activated partial thromboplastin time (aPTT) were determined with the reagents Hepato Prest and STA aPTT-A, respectively (Diagnostica Stago, Asnieres, France). The main outcome parameter was the blood concentration of DOACs, which was determined in plasma obtained from citrated venous blood immediately before (trough level) and two to three hours after drug intake (peak level). DOACs were administered in the presence of the treating physician, and rivaroxaban was taken together with food. Drug concentrations were determined by chromogenic anti-Xa assays calibrated for apixaban, edoxaban, or rivaroxaban (STA Liquid Anti-Xa, Diagnostica Stago, Asnieres, France) or by an Ecarin chromogenic assay for dabigatran (STA-ECA II, Diagnostica Stago). Respective calibrators and controls were obtained from Diagnostica Stago. Expected ranges for trough and peak levels differed depending on the DOAC and the respective dosage and indication, and were adopted according to the “International Council for Standardization in Haematology Recommendations for Laboratory Measurement of Direct Oral Anticoagulants” [20].

All coagulation assays were performed fully automated on a STA-R Max2 analyzer (Diagnostica Stago).

### 2.3. Statistical Analysis

Statistical analysis was performed with the Statistical Package for the Social Sciences (SPSS) 20.0 (SPSS Inc., Chicago, IL, USA). Nominal and ordinal scaled parameters were described with absolute and relative frequencies. Metric variables are presented as median and interquartile ranges (IQR).

## 3. Results

### 3.1. Patient Characteristics

Ten of the eleven patients had gastroesophageal cancer, and one patient had pancreatic cancer. Eight patients had total and three patients had partial gastrectomy. Patient characteristics are provided in Table 1. Their median age was 76 years (IQR 56–78), and eight (73%) were men.

Ten patients were treated with an anti-Xa inhibitor (three with apixaban, three with edoxaban, and four with rivaroxaban), and one patient received dabigatran at the time of referral to the Division of Hematology and Hemostaseology.

Seven patients were treated for a VTE, which occurred around the time of gastrectomy in five patients and two years after gastrectomy in two patients. In all patients, DOACs were started upon diagnosis of VTE.

Four patients had nonvalvular atrial fibrillation, which had been diagnosed two years after gastrectomy in two patients and one year before gastrectomy in the other two patients. Three of these patients were initially treated with DOACs. Patient 9 was initially treated with a vitamin K antagonist. After an intracerebral hemorrhage, anticoagulation was switched to apixaban.

The median interval between the start of DOAC treatment after gastrectomy and day of concentration measurement was one month (IQR 0.9–9). The median time from first intake of DOACs after gastrectomy until the last record in the medical charts, until discontinuation of anticoagulation or death, was 10 months (IQR 8–22).

### 3.2. DOAC Concentrations and Patient Management

Information on anticoagulant treatment and coagulation parameters are provided in Table 2. All except two of the ten patients receiving a factor Xa inhibitor had anti-Xa trough levels within the expected range. Two to three hours after drug intake, levels increased in all patients, albeit to variable degrees.

Patients 2 and 3 received edoxaban at the lower dose of 30 mg due to a body weight ≤60 kg, and both had trough levels below the detection limit. Patient 3 was switched to rivaroxaban 20 mg, which resulted in trough levels in the expected range, whereas peak levels were still below the expected range. Considering trough levels in the expected range, treatment with rivaroxaban was maintained without thromboembolic or bleeding complications during follow-up. Patient 8 discontinued edoxaban and refused further anticoagulation. 

DOAC concentrations did not markedly differ between patients with total or partial gastrectomy. 

Four patients (patients 3, 7, 8, and 10) received pharmacologic antineoplastic therapy at the time of DOAC concentration measurement, and had concentrations which were comparable to those of patients without anticancer treatment. 

INR and the activated partial thromboplastin time (APTT) were within the normal range or only marginally altered at trough levels, and did not substantially change at peak DOAC concentrations.

### 3.3. Thrombotic and Bleeding Events

None of the patients had a thromboembolic event during treatment with a factor Xa inhibitor. Patient 1 had major bleeding after a brachial artery puncture that was successfully treated with compression. Once adequate hemostasis had been achieved, the patient continued anticoagulation with edoxaban without further bleeding. Patient 10 had gastrointestinal bleeding during local irradiation. Apixaban was discontinued and not restarted within a follow-up limited to four weeks.

Patient 11 had nonvalvular atrial fibrillation (CHA_2_DS_2_-VASc score: 2) and an ischemic stroke during thromboprophylaxis with dabigatran at a dose of 110 mg twice daily. Trough and peak dabigatran concentrations were below the expected range (Table 2). The dabigatran dose was increased to 150 mg twice daily without a substantial effect on trough and peak concentrations. Shortly before switching to a different DOAC, the patient had a second ischemic stroke. He was then switched to apixaban, which led to trough and peak anti-Xa levels within the expected range. During the following 10 months, the patient neither had thromboembolic nor bleeding events.

## 4. Discussion

The principal finding of our study was that patients with gastrointestinal cancer, who had undergone partial or total gastrectomy and were treated with a direct oral factor Xa inhibitor, achieved plasma concentrations that were within the expected range. Our findings therefore indicated that DOACs, particularly the factor Xa inhibitors, were adequately absorbed despite pharmacokinetic data indicating that maximum plasma concentrations were decreased when the drugs were released in the small intestines or ascending colon [22,23], rather than in the stomach or more proximal parts of the intestine.

Rivaroxaban is absorbed predominantly in the stomach, and due to its lipophilicity must be taken with food [24]. It was thus unexpected that four out of five patients who had undergone gastrectomy had rivaroxaban trough and peak levels well within the expected range. Our findings of an adequate absorption of rivaroxaban in cancer patients after gastrectomy were supported by Mahlmann et al. who reported a bariatric surgery patient with a rivaroxaban peak concentration in the expected range [15]. In contrast, five of seven bariatric surgery patients had significantly lower peak concentrations of rivaroxaban compared to matched controls [14]. Interestingly, in the same study, 11 patients had an adequate increase in their respective plasma concentrations after apixaban (9 patients) or dabigatran (2 patients) [14]. It is unclear, however, whether observations made in bariatric surgery patients can be extrapolated to patients after gastrectomy for cancer, as the two populations differ from each other in important patient characteristics, including type of surgery, age, body weight, and presence of comorbidities. 

Apixaban is characterized by good water solubility, and its bioavailability is not affected by food intake. Apixaban is absorbed throughout the gastrointestinal tract, with more than 50% of its absorption in the distal small bowel and the ascending colon, and also in the more distal parts of the intestines [25]. All four of our patients who were treated with apixaban achieved both trough and peak apixaban plasma concentrations in the expected range. This finding was in line with a study in bariatric surgery patients reporting apixaban plasma concentrations in the expected range in all nine patients receiving apixaban [14]. The pharmacokinetic characteristics of apixaban, together with our findings, suggest that apixaban also could be advantageous for cancer patients after gastrectomy.

Two of our patients received edoxaban at a dose of 30 mg once daily because their body weight was less than 60 kg. In both patients, edoxaban trough levels were below the detection limit. Considering the short half-life of edoxaban of 10 hours, it could well be that the lower edoxaban dose might not have been high enough in this specific patient population. Edoxaban peak levels were, however, in the expected range, precluding reduced bioavailability and contrasting data that edoxaban is most soluble in the acidic environment of the stomach [23].

Dabigatran is administered as an etexilate ester prodrug, and is more consistently absorbed in an acidic milieu in the gastrointestinal tract. It is formulated together with tartaric acid, which makes it independent of the gastrointestinal acidity, suggesting that its absorption is not restricted to the stomach [26]. Importantly, the absolute bioavailability after oral administration of dabigatran etexilate is only ≈7%, requiring administration of higher doses to ensure adequate plasma concentrations. In our study, we encountered only a single patient who was treated with dabigatran for atrial fibrillation after gastrectomy for stomach cancer. This patient had an embolic stroke while on dabigatran at a dose of 110 mg twice daily, and suffered from recurrence after the dabigatran dose had been increased to 150 mg twice daily. Trough and peak dabigatran plasma concentrations were well below the expected range after both the low and high doses. In a prespecified analysis of the Randomized Evaluation of Long-Term Anticoagulation Therapy (RE-LY) trial, the risk of ischemic events among patients with atrial fibrillation was inversely related to steady-state trough dabigatran concentrations [27]. Considering that the patient had a CHA_2_DS_2_-VASc score of only 2, it stands to reason that dabigatran plasma levels were inadequate to prevent arterial thromboembolism, rather than a breakthrough event due to an excessively high thrombotic risk. The patient was later switched to apixaban, which resulted in both adequate trough and peak apixaban plasma levels and prevention of stroke during an observation period of 10 months. This observation indicated that routine measurement of blood DOAC levels in patients after gastrectomy may help to individualize the optimal anticoagulation strategy.

There were only few reports in the literature on patients treated with dabigatran after gastrectomy and there was only one on a cancer patient. Bolek et al. reported a patient with diffuse large B-cell lymphoma who underwent total gastrectomy and was treated with dabigatran for atrial fibrillation. He had low dabigatran trough and peak levels and was switched to apixaban, resulting in adequate apixaban plasma concentrations at both trough and peak [19]. In 2013, Lee et al. reported two patients with atrial fibrillation who had subtherapeutic dabigatran levels after gastric bypass surgery [17]. Lachant et al. reported a patient after bariatric gastric bypass surgery who had a pulmonary embolism despite anticoagulation with dabigatran for concomitant atrial fibrillation [12]. The authors did not measure dabigatran plasma concentrations, but inferred from a normal APTT that the absorption of dabigatran might have been insufficient. In a retrospective case series of nine patients who underwent gastric bypass surgery, dabigatran peak concentrations were consistently lower than those reported in clinical trials [28]. Five of these nine patients were switched to rivaroxaban, and peak serum concentrations were comparable to the range reported in the literature. 

DOACs affect global coagulation assays, including INR and APTT. Factor Xa inhibitors more strongly increase the INR, while the thrombin inhibitor dabigatran prolongs the APTT, and to a lesser extent increases the INR. In our study, APTTs and INRs measured at DOAC trough concentrations were normal or only mildly prolonged. At DOAC peak concentrations, test results were variable and not much different from trough levels. Thus, in cancer patients who underwent gastrectomy, global coagulation assays are not informative and do not reliably reflect the impact of DOACs on the clotting system.

Some strengths and limitations need to be addressed. Our study described the largest cohort of cancer patients who underwent gastrectomy and concomitantly received antithrombotic treatment with a DOAC so far. Nevertheless, the limited number of patients precluded more than a descriptive data analysis. Considering the increasing evidence that DOACs are safe and effective in the general cancer population, this information is pertinent in order to improve care of cancer patients after gastrectomy and to address demands of patients and health care providers. DOAC concentrations were obtained according to defined standards with accredited assay systems. We measured DOAC concentrations only once in each patient, since we aimed at assessing the absorption rates rather than monitoring drug levels. We used therapeutic ranges obtained from published studies as a surrogate for DOAC effectiveness instead of clinical endpoints [20,21]. 

## 5. Conclusions

Direct oral factor Xa inhibitors were adequately absorbed in cancer patients after partial or total gastrectomy, suggesting that their use in clinical practice might be justified. Intermittent assessment of trough and peak levels should be considered to assure adequate absorption. In a single patient, both trough and peak plasma concentrations after administration of the thrombin inhibitor dabigatran were well below the expected range both after the low and the high dose, and he suffered from recurrent strokes. Therefore, more data on the pharmacokinetics of dabigatran in this specific patient population are needed, and other patients should be studied due to the potential high variability of individual responses. Until then, dabigatran should be used with caution in cancer patients after gastrectomy.

## Figures and Tables

**Table 1 pharmaceutics-14-00662-t001:** Patient characteristics.

Patient Number	Sex	Age (years)	BMI (kg/m^2^)	Weight (kg)	Cancer (Type and Stage)	Gastrectomy	Indication for Anticoagulation	Time before DOAC Measurement (mo) *	Observation Time (mo) **
1	m	80	28	88	GEJ adenocarcinoma–stage I	Total	AF	7	17
2	f	81	17	39	Gastric adenocarcinoma–stage II	Total	AF	0.1	0.1
3	f	37	18	43	Pancreatic adenocarcinoma–stage IV	Total	DVT, PE	1	7
4	f	55	26	70	Gastric adenocarcinoma–stage I	Total	PE	1	8
5	m	57	20	65	GEJ adenocarcinoma–stage III	Total	Abdominal vein thrombosis	10	23
6	m	76	27	84	GEJ adenocarcinoma–stage III	Partial	PE	8	20
7	m	76	28	92	GEJ adenocarcinoma–stage III	Total	DVT, PE	1	10
8	m	76	22	65	GEJ adenocarcinoma–stage II	Partial	DVT	22	34
9	m	81	23	73	Gastric neuroendocrine carcinoma–stage III	Total	AF	0.2	9
10	m	55	20	64	GEJ adenocarcinoma–stage III	Partial	DVT	0.7	6
11	m	73	23	70	GEJ adenocarcinoma–stage III	Total	AF	62	70

GEJ–gastroesophageal junction, PE–pulmonary embolism, DVT–deep vein thrombosis of the leg, AF–non-valvular atrial fibrillation. * Time between start of DOAC and DOAC level measurement. ** Time between first intake of DOAC after gastrectomy and last record in medical charts, discontinuation of anticoagulation or death.

**Table 2 pharmaceutics-14-00662-t002:** Trough and peak DOAC concentrations and coagulation parameters.

				Trough Level				Peak Level			
Patient Number	DOAC	Dosage (mg)	Creatinine Clearance (mL/min)	DOAC Level (ng/mL)	Expected Range * (ng/mL)	PT INR	aPTT (s)	DOAC Level (ng/mL)	Expected Range * (ng/mL)	PT INR	aPTT (s)
1	Edoxaban	1 × 60	78	51	19–62	1.1	39.5	195	125–245	1.2	47.7
2	Edoxaban	1 × 30	40	n.d.	4–20	1.2	40.4	171	60–120	1.3	49.9
3A	Edoxaban	1 × 30	68	n.d.	4–20	1.2	28.8	83	60–120	1.3	33.6
3B	Rivaroxaban	1 × 20	87	22	6–87	1.2	29.9	39	189–419	1.2	31.2
4	Rivaroxaban	1 × 20	88	45	6–87	1.0	34.1	210	189–419	1.0	44.3
5	Rivaroxaban	1 × 20	94	27	6–87	1.0	30.4	437	189-419	1.1	39.4
6	Rivaroxaban	1 × 20	68	65	6–87	1.3	34.5	257	189–419	1.4	43
7	Rivaroxaban	1 × 20	55	40	6–87	1.1	32.8	228	189–419	1.2	40.8
8	Apixaban	2 × 2.5	58	26	11–90	1.0	35	65	30–153	1.1	38
9	Apixaban	2 × 2.5	41	45	34–162	1.0	36.3	126	69–221	1.1	38.6
10	Apixaban	2 × 5	99	92	22–177	1.0	31.3	261	59–302	1.0	33.6
11A	Dabigatran	2 × 110	64	18	28–155	1.0	31.5	53	52–275	1.0	37.9
11B	Dabigatran	2 × 150	61	21	61–143	1.1	36.2	46	117–275	1.1	40.4
11C	Apixaban	2 × 5	63	99	41–230	1.1	33.5	217	91–321	1.1	35.3

Abbreviations: n.d.–not detectable, PT INR–prothrombin time based international normalized ratio, aPTT–activated partial thromboplastin time (reference range 27.0–41.0 seconds), creatinine clearance estimated by the Cockcroft-Gault equation. * Expected ranges differ depending on the DOAC and the respective dosage and indication [20].

## Data Availability

The data that support the findings of this study are available from the corresponding author, L.E., upon reasonable request.

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
