# Peer review of "Absorption of Direct Oral Anticoagulants in Cancer Patients after Gastrectomy"

_pharmaceutics, 2022, doi:10.3390/pharmaceutics14030662_

Round 1

Reviewer 1 Report

I have read with interest the paper by Puhr et al entitled “Absorption of direct oral anticoagulants in cancer patients after gastrectomy”. This study is interesting because it gives information on a condition which can be encountered in the daily clinical practise. I have only a few minor points to be addressed to the authors.

1 Prothrombin Time must not be reported as percentage. PT INR is appropriate.

2 The expected ranges for DOAC should be made in the own laboratory. This because the variability of the single person is high. Local behaviour of people should always be considered.

3 Dabigatran can also prolong Prothrombin Time, not only aPTT.

4 To conclude against the use of Dabigatran in patients submitted to gastrectomy is not correct because the analysis of only one patient is not enough. Other patients sould be studied since the high variability of the individual response.  This statment should be added to the text. 

Author Response

Please find our responses in the file attached.

Reviewer 2 Report

See file attached

Author Response

(The authors gave the same response as above.)
